

# Extracting real-crack properties from nonlinear elastic behavior of rocks: abundance of cracks with dominating normal compliance and rocks with negative Poisson's ratio

Vladimir Y. Zaitsev[1], Andrey V. Radostin[1], Elena Pasternak[2], and Arcady Dyskin[3]

[1]Institute of Applied Physics, Russian Academy of Sciences, Nizhny Novgorod, 603950, Russia
[2]School of Mechanical and Chemical Engineering, The University of Western Australia, Perth, 6009, Australia
[3]School of Civil, Environmental and Mining Engineering, The University of Western Australia, Perth, 6009, Australia

Correspondence to: Vladimir Y. Zaitsev (vyuzai@ipfran.ru)

**Abstract.** Results of examination of experimental data on nonlinear elasticity of rocks using experimental pressure-dependences of P- and S-wave velocities from various literature sources are presented. Overall, over 90 rock samples are considered. Interpretation of the data is performed using an effective-medium description in which cracks are considered as compliant defects (cracks) with independent shear and normal compliances without specifying a particular crack model with an *a priori* given ratio of the compliances. Comparison with the experimental data indicated abundance of cracks (~80%) with the normal-to-shear compliance ratios significantly exceeding the values typical of conventionally used crack models (such as penny-shape cuts or thin ellipsoidal cracks). Correspondingly, rocks with such cracks demonstrate strongly decreased Poisson's ratio including a significant portion of rocks (~45%) exhibiting negative Poisson's ratios at lower pressures, for which the concentration of not yet closed cracks is maximal. The obtained results indicate the necessity of further development of crack models to account the revealed numerous examples of cracks with strong domination of normal compliance. Discovering such a significant number of naturally auxetic rocks is in contrast with the conventional viewpoint that occurrence of negative Poisson's ratio is an exotic fact that is mostly associated with specially engineered structures.

## 1 Introduction

It is widely appreciated that most rocks exhibit strongly increased tensosensitivity, that is giant elastic nonlinearity as compared with atomic nonlinearity of homogeneous solids and liquids. A bright manifestation of this nonlinearity is a very pronounced dependence of rocks' elastic moduli on applied pressure. The main reasons for this giant nonlinearity is the presence of highly compliant cracks and contacts in the relatively hard matrix.

Important features of this "soft-hard paradigm" of giant nonlinearity in microstructured solids [1,2] can be explained by very instructive and simple 1D rheological models in which highly-compliant cracks/contacts correspond to soft elastic elements/springs contained in a relatively hard matrix [3,4,5]. Such models can be very useful to elucidate as to why the relationship between concentration of the soft inclusions and the resultant nonlinearity level can be non-monotonic Also they





can provide some understanding of the origin of frequency dependence of such microstructure-induced nonlinearity as an influence of relaxation localised at the same soft defects. Furthermore, those rheological models clearly demonstrate that the relaxation properties of the soft defects in addition to the elastic nonlinearity (i.e. tensosensitivity of elastic moduli) inevitably lead to pronounced tensosensitivity of dissipation in microstructured solids [6,7] that may exhibit itself as

dissipative nonlinearity.

Despite usefulness of the above-mentioned 1D models for understanding basic features of the influence of high-compliant inclusions on reduction of the elastic modulus and the origin of its giant stress-sensitivity, closer comparison with seismo-acoustic properties of real rocks require the effective-medium models that more adequately correspond to a 3D character of real rocks. Even in the simplest isotropic approximation, rocks are characterised by two independent elastic moduli. The

most widely used are the bulk modulus, shear modulus determining the velocity of shear S-waves, Young modulus, as well as the modulus corresponding to the velocity of longitudinal P-waves. Among those moduli any two are independent and the other are expressed via the chosen pair of the independent ones.

Since cracks are the simplest and most important type of compliant defects in consolidated rocks, considerable attention was paid to developing models that describe crack-induced variations in elastic moduli. Although such descriptions differ in the

way of accounting for eventual interaction of cracks (i.e. small-perturbation or approximation of low crack concentrations, without accounting for mutual crack interaction [8], the so-called self-consistent approach [9] or differential approach [10]), the representations of cracks in such models were based on simpe geometries, for which exact expressions were available; these describe the stored elastic energy in the presence of shear stress or stress normally directed to the crack plane. In particular, the so-called penny-shape cracks or thin elliptical voids with small aspect ratios have been widely used.

Despite the differences in the methods accounting for interaction of cracks at larger concentrations, in the limiting case of small crack concentrations all of such models predict identical complementary variations for the chosen independent elastic moduli. For example, the chosen crack geometry pre-determines a given very specific proportion between variations in the S- and P-wave velocities under hydrostatic pressure. Observations for real rocks, however, often demonstrate different proportions between crack-induced variations in the P- and S-wave velocities variations, such that playing with crack

concentrations in the above-mentioned models in principle cannot help to reach better agreement between the predictions and observations.

The fact that variations of moduli inferred from the measured wave velocities require different crack concentrations for different moduli (e.g., different concentrations to obtain the values of $E$ and $G$ inferred from the wave velocities), implies that real cracks could be characterised by significantly different proportions between their shear and normal compliances.

Such variability of crack properties in principle cannot be accounted for in conventional effective-medium models based on cracks modelled as straight cuts of any geometry (e.g., penny-shape) or thin ellipsoidal voids with a small aspect ratio. In such conventionally used models the ratio between those compliances is pre-determined and cannot exhibit significant variations.





This circumstance motivated the development of alternative effective-medium models in which cracks are considered as highly compliant defects with independent normal and shear compliances not restricted by a predetermined proportion between them. Such an idea was realized in [11] and equivalent expressions (that differ only by a normalization) were derived in [12] based on results of [13]. Using results [11], the ratios between normal and shear compliances were extracted

in [14] from the analysis of pressure dependences of two elastic-wave velocities in three samples. For one of the samples, the inferred crack characteristics did not differ strongly from the ones obtained using the conventional penny-shape crack models, whereas the other two demonstrated 2-4 times stronger dominance of normal compliance of the real defects. Furthermore, one of the samples (Weber sandstone studied in [15]) with the highest normal-to-shear compliance ratio of the cracks was found to possess negative Poisson's ratio at lower confining pressures (up to 20 MPa). With increasing pressure

(that caused gradual closing of the cracks) the Poisson's ratio gradually increased towards to the "normal" positive values.
Results [14] demonstrated that properties of real cracks may significantly differ from those implied in the popular model of penny-shape cracks. This agrees with some recent works [16,17] where some other facts indicating insufficiency of models based on penny-shape cracks are discussed. However, the fairly small number of rocks discussed in those papers did not yet allow one to estimate how exotic are samples where the pressure dependence of the moduli is inconsistent with the models

based on conventional cracks. In what follows, we present results of the examination of pressure dependences for ~90 rocks [18,19,20] demonstrating that the "unusual" properties of real cracks are quite common. Furthermore, we show that in contrast to the common belief the relevance of the conventional crack concept can be considered as an exception, while the rocks with negative Poisson's ratio are not rare.. Reliable reconstruction of compliance properties of cracks (that are conventionally used in models of linear elastic properties of rocks) requires consideration of nonlinear behaviour of rocks –

the pressure-induced variation of their elastic properties – in a sufficiently wide pressure range. In the course of this consideration we will also point out some aspects of rock's nonlinearity (tensosensitivity) that have not been explicitly discussed earlier.

## 2 Nonlinear variation of rock's elasticity under varying pressure: 1D modelling

In geophysics elastic nonlinearity of rocks is well appreciated, however when considering nonlinear propagation of elastic

waves the modelling is often simplified by using 1D approximation starting from 1D constitutive nonlinear stress-strain relationship in which quadratic in strain nonlinearity is often considered. For the present consideration of nonlinear variations of elastic moduli under isotropic hydrostatic compression that affects the state high-compliant defects, the 1D description can also be used, for example, in the form with small (quadratic in strain) nonlinear correction to the linear stress-stress relationship:

$$\sigma(\varepsilon) = \varepsilon \cdot M \ \{1 + \varepsilon \cdot \gamma^{(2)}\} = \varepsilon \cdot M \ + \gamma^{(2)} \cdot M \ \cdot \varepsilon^2 \ , \tag{1}$$

Here $M$ is the effective modulus of the medium and $\gamma^{(2)}$ is a dimensionless nonlinearity parameter characterizing variability of the elastic modulus with variations in strain acting in the medium. Strictly speaking, strain is defined with

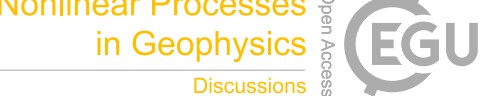



respect to some initial state of the medium, so that the local slope of the dependence around a current degree of deformation becomes dependent on strain (and thus stress) and can be considered as an incremental elastic modulus:

$$M(\varepsilon) = \partial\sigma/\partial\varepsilon \qquad (2)$$

It is clear that the dimensionless parameter of the quadratic nonlinearity can be expressed as

$$\gamma_{eff}^{(2)} = \frac{1}{2 \cdot M}\partial M(\varepsilon)/\partial\varepsilon = \frac{1}{2}\frac{dM}{d\sigma} \qquad (3)$$

taking into account that the stress (pressure) increment is related to the strain increment as $d\sigma = M_{eff} \cdot d\varepsilon$.

In contrast to homogeneous materials with weak atomic nonlinearity and the nonlinearity parameter on order of unity [21,1,2], in heterogeneous media their nonlinearity can be strongly increased due to the presence of highly compliant defects with strongly locally decreased elastic modulus. Due to this fact the strain becomes strongly locally increased at the soft defects, which results in a considerable enhancement of their local nonlinear deviation from the linear stress-strain law and, correspondingly, leads to enhancement of average (macroscopic) nonlinearity of the material.

Important features of the microstructure-related nonlinearity can be revealed in the framework of the above-mentioned 1D description [3,4]. The simplest for understanding is the case of identical compliant defects: if the relative volume content (concentration) of such highly compliant defects is $\upsilon$ and the defects are of the same type, the effective quadratic nonlinearity parameter gets strongly increased

$$\gamma^{(2)}/\gamma_0^{(2)} \approx (1+\upsilon/\varsigma^2)\big/(1+\upsilon/\varsigma)^2 \qquad (4)$$

where the small parameter, $\varsigma \ll 1$, characterizes the relative compliance of the compliant defects with respect to the homogeneous matrix material; parameter $\gamma_0^{(2)} \sim 1$ characterizes the own weak nonlinearity of the material of the defects. A clear example is a liquid with gas bubbles: taken separately the liquid and gas both are weakly nonlinear, but the nonlinearity of the mixture may become giant. For sufficiently small compliance parameter $\varsigma \ll 1$ (that may be $10^{-3} - 10^{-4}$ for gas-water mixture), the nonlinearity parameter can exhibit giant increase, $\gamma_{eff}^{(2)}/\gamma^{(2)} \gg 1$, even for small concentrations $\upsilon$, because the combination $\upsilon/\varsigma^2$ in Eq. (4) may become large. Simultaneously with the increase in the nonlinearity parameter, the elastic modulus $M$ due to the presence of high compliant defects exhibits gradual decrease in comparison with modulus $M_0$ of the homogeneous matrix:

$$M / M_0 \approx 1/(1+\upsilon/\varsigma), \qquad (5)$$

Comparing Eqs. (3) and (4) one can easily notice that even if the decrease in the elastic modulus is small, $\upsilon/\varsigma \ll 1$, the increase in the nonlinearity parameter, Eq. (4) may become significant ($\gamma_{eff}^{(2)}/\gamma^{(2)} \gg 1$), since the combination $\upsilon/\varsigma^2$ may become large even if $\upsilon/\varsigma \ll 1$. Furthermore, the nonlinearity parameter reaches its maximum value $\sim 1/\varsigma$ for rather small





concentration of the defects $\upsilon = \varsigma$, for which the elastic modulus decreases twice and the interplay between the local strain enhancement and the concentration of the defects is optimal [3,4].

In contrast to the above-mentioned bubbly liquids, for which the bubbles have the same contrast $\varsigma$ in compressibility relative to the liquid where the existence of clear maximum of the nonlinear parameter in its dependence on bubble

concentration is a known fact [22,23], for cracked rocks, the presence of maximum nonlinearity at an intermediate concentration of cracks is not typical. Bearing in mind that for the bulk modulus $K$ of rocks under hydrostatic compression, the 1D description is applicable as far as the normal compliance of cracks is concerned (see below for more details), we note that pressure dependences of $K(P)$ usually demonstrate ever increasing slope $dK_{eff}/dP$ (i.e., the nonlinearity parameter) with reducing confining pressures $P$ at which the concentration of cracks that are not closed gradually increases. Typical

examples of $K_{eff}(P)$ recalculated from experimentally measured P- and S-wave velocities are shown in Fig. 1a for several sandstone samples that are often discussed in literature [15,19].

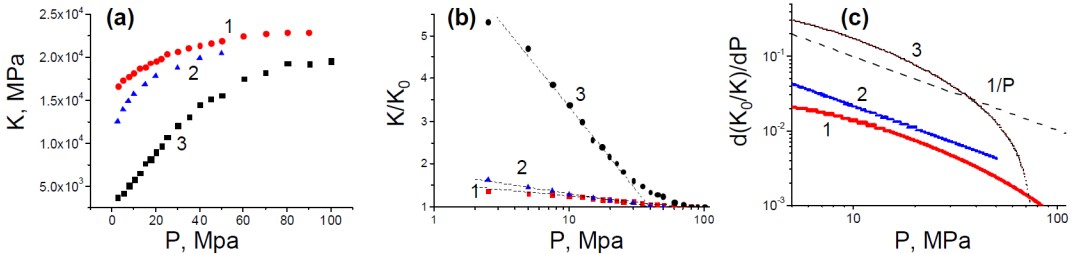

**Fig. 1. Non-linearity exhibited by dry Navajo, Nugget and Weber sandstones [15,19]: (a) typical dependences $K(P)$ recovered from the experimentally measured pressure-dependences of P- and S-wave velocities; (b) the same data**

**represented as the pressure dependences for inverse bulk modulus $K^{-1}(P)$ shown in the plot with logarithmic pressure-axis; (c) derivatives $d(1/K)/dP = -(dK/dP)/K^2$ calculated using the approximating curves (in the form of 3rd order polynomials). Numbers 1, 2 and 3 denote the data for Navajo, Nugget and Weber sandstones, respectively [15,19,20]. The slopes of the approximating straight lines corresponding to normalized Eq. (8) characterize the differences in the density of cracks ($\nu_0^{(\varsigma)} \approx 0.13$ for Navajo, $\nu_0^{(\varsigma)} \approx 0.21$ for Nugget and $\nu_0^{(\varsigma)} = 1.7$ for Weber). The**

**dashed line in panel (c) shows the $1/P$ dependence corresponding to the normalized Eq. (10). The deviation downwards of curve 3 for Weber in panel (c) is related to pronounced saturation of $K(P)$ and $K^{-1}(P)$ at higher pressures clearly visible in panels (a) and (b).**




This gradual increase in slope $dK/dP$ with decreasing pressure is quite naturally attributed to a broad distribution of the compliant defects over their compliance parameter. Indeed, it is widely accepted that with increasing confining pressure the compliant crack-like defects gradually become tightly closed (starting from the most compliant) and do not contribute anymore to the rock nonlinearity. This agrees with the known properties of narrow cracks with small aspect ratio $\alpha \ll 1$, for

which their aspect ratio determines the relative compliance $\alpha \sim \varsigma$. Such thin cracks are known to get closed under the average strain $\varepsilon_c \sim \alpha$; the proportionality coefficient is of order of unity and its value may somewhat differ as demonstrated by the solutions for elliptical cracks [24], tapered non-elliptical cracks [25], etc. Since the strains and applied pressure $P$ can be considered roughly proportional, $\varepsilon_c \approx P_c/K$, all these quantities can be considered as being approximately proportional to each other: $\varsigma \sim \alpha \sim \varepsilon_c \sim P_c/K$; this will be taken into account in the consideration below.

For not-identical defects with a distribution in the compliance parameter, Eqs. (3) and (4) should be modified to comprise the contributions of defects with different compliance parameter $\varsigma$ [5]:

$$K/K_0 \approx 1/(1+\int \frac{\upsilon(\varsigma)}{\varsigma} d\varsigma) \tag{6}$$

The equation for the nonlinear parameter can be rewritten as

$$\frac{K_0}{K^2}\gamma^{(2)}/\gamma_0^{(2)} \approx 1+\int \frac{\upsilon(\varsigma)}{\varsigma^2} d\varsigma \tag{7}$$

It is clear that, by analogy with Eqs. (4) and (5) for identical defects, the modulus reduction and the increase in nonlinearity are determined by the distribution $\upsilon(\varsigma)$ of defect concentration over the compliance parameter $\varsigma$.

If one consider ranges of pressure $P_{min} \le P \le P_{max}$ relevant to experiments, quite often this range is from several MPa to about $10^2$ MPa, i.e. with relative variation $P_{max}/P_{min} \sim 15-30$ times, as the examples in Fig. 1 show, the gradually closed/opened cracks should be distributed over the compliance parameter with a similar relative range,

$\varsigma_{max}/\varsigma_{min} \sim P_{max}/P_{min}$. Since this range in practically relevant cases is not huge (not many orders of magnitude), one can assume that in the first approximation the function $\upsilon(\varsigma)$ may be approximated by a uniform distribution, $\upsilon(\varsigma)=\upsilon_0^{(\varsigma)} \approx const.$ for $\varsigma_{min} \le \varsigma \le \varsigma_{max}$. Then one obtains:

$$K_0/K \approx 1+\int_{\varsigma}^{\varsigma_{max}} \frac{\upsilon(\varsigma)}{\varsigma} d\varsigma = 1+\upsilon_0^{(\varsigma)}\ln(\varsigma_{max}/\varsigma) \approx 1+\upsilon_0^{(\varsigma)}\ln(P_{max}/P) \tag{8}$$

$$\frac{K_0}{K^2}\gamma^{(2)}/\gamma_0^{(2)} \approx 1+\int_{\varsigma_{min}}^{\varsigma_{max}} \frac{\upsilon(\varsigma)}{\varsigma^2} d\varsigma = 1-\frac{\upsilon_0^{(\varsigma)}}{\varsigma_{max}}+\frac{\upsilon_0^{(\varsigma)}}{\varsigma} \tag{9}$$





Practically more useful than Eq. (9) can be a representation in the form of the direct derivative of Eq. (8), $d(1/K)/dP = -(dK/dP)/K^2$, that does not involve unknown initial value of the nonlinearity parameter. In view of relationship (8) it should be expected in the form

$$\frac{K_0}{K^2}\frac{dK}{dP} \approx \frac{\upsilon_0^{(\varsigma)}}{P} \tag{10}$$

This dependence can be compared with experimental data. Figure 1b shows the pressure dependences for the bulk modulus of the same samples as in Fig. 1a using logarithmic scale of the pressure axis, for which proportionality to $\log(P)$ should look as a straight line. It is clear that in Fig. 1b such straight lines approximate the experimental dependences $K^{-1}(P)$ fairly well. The trends to saturation closer to maximal and minimal strains are expectable (since the distribution $\upsilon(\varsigma) = \upsilon_0^{(\varsigma)} \approx const.$ cannot be ideally flat). The slopes of the straight lines in Fig. 1b are determined by $\upsilon_0^{(\varsigma)}$ and give clear

representation on the differences in the characteristic concentrations of the defects for the examined samples. Finally, Fig. 1c presents, in log-log scale, the derivatives of the approximating curves shown in Fig.1b with a dependence $1/P$ as a guide. Thus Figs. 1b and 1c demonstrate that the simplest approximation of the distribution of the defects by a constant value reasonably agrees with the experimental observations in fairly wide range of pressures ($P_{\max}/P_{\min} \sim 10-20$ times).

### 3 Inferences from nonlinear variations in elastic moduli of rocks in 3D description

In the previous section we considered only 1D description that can be quite well applied to the reduction in the bulk modulus under hydrostatic compression of real rock samples. However, in real 3D rocks even under isotropic hydrostatic compression and fairly isotropically oriented cracks, there exist two independent elastic moduli of which the bulk modulus and shear modulus are often considered. The crack-like defects with isotropic orientations can also be characterized by two independent compliances with respect to normal and shear loading. Using such a representation of cracks like planar defects

with two compliances that are not *a priori* predetermined by a particular crack model one can relate the values of different elastic moduli with the crack effective densities and compliances by analogy with the above considered 1D case. Such expressions were obtained in [11] in the form

$$\tilde{K} = \frac{K_{eff.}}{K} = \frac{1}{1 + \frac{1}{3}N_n/(1-2\nu)} \tag{11}$$

$$\tilde{G} = \frac{G_{eff.}}{G} = \frac{1}{1 + \frac{2}{15}N_n/(1+\nu) + \frac{2}{5}N_s} \tag{12}$$

where by analogy with the above-considered 1D case, $N_1 = \int \upsilon(\varsigma)\varsigma^{-1}d\varsigma$ is the effective concentration of the normal compliance and $N_2 = \int \upsilon(\xi)\xi^{-1}d\xi$ is a similar quantity for the shear compliance, and $\nu$ is the Poisson's ratio of the matrix





rock. For the other moduli, one obtains similar expressions [11]. In these equations the shear compliance is normalized by the shear modulus of the rock matrix and the normal compliance is normalized by the Young modulus corresponding to the rock deformability under uniaxial stress. Instead of a single dimensionless compliance parameter used in the previous section (in fact corresponding to the normal compliance) these expressions contain two compliance parameters representing the

normal and shear loading characterize the defects. Factors 1/3, 2/5, etc. in Eqs. (11) and (12) are related to spatial averaging of isotropically oriented defects.

Similar equations were derived in [12] using basic relations obtained in [13]:

$$\tilde{K} = \frac{K_{eff.}}{K} = \frac{1}{1 + K_0 Z_n} \tag{13}$$

$$\tilde{G} = \frac{G_{eff.}}{G} = \frac{1}{1 + \frac{4}{15} G_0 Z_n + \frac{2}{5} G_0 Z_s} \tag{14}$$

In these equations quantities $Z_1$ and $Z_2$ characterizing total normal and shear compliances imparted to the rock by cracks are dimensional (with dimension of inverse modulus). The shear compliance of the defects in both approaches (i.e., Eqs(12) and (14)) is similarly compared with the shear elastic modulus of the matrix material. However, the normal compliance in Eq. (12) is normalized differently: in [11], the normal compliance of the defects is compared with the Young modulus (i.e. the modulus that corresponds to uniaxial stress, so that the compliance parameter $\varsigma$ of the defects with respect to normal

uniaxial stress can be expressed as $\varsigma = E_{crack} / E_0$, the latter can be substituted in the expression for $N_n = \int \nu(\varsigma) \varsigma^{-1} d\varsigma$). Then taking into account conventional relationship $K = \frac{1}{3} E / (1 - 2\nu)$ between moduli $E$ and $K$ [26], the combination $\frac{1}{3} N_n / (1 - 2\nu)$ in Eq. (11) can be transformed into the form $\frac{1}{3} N_n / (1 - 2\gamma) = K_0 Z_n$ (where $Z_n = \int \nu(\varsigma) \varsigma^{-1} d\varsigma / E_0$). As a result, Eq. (11) assumes the form of Eq. (13) in the notations of paper [12], where the normal compliance of the defects is normalized using the bulk modulus $K_0$ of the matrix. This comparison justifies that Eqs. (11) and (13) for the effective bulk

modulus have the same form as the one-dimensional Eq. (6) discussed in the previous section.

Note further that the total shear compliances $\frac{2}{5} N_s$ and $\frac{2}{5} G_0 Z_s$ in Eq. (12) and (14) have exactly the same meaning (coincide quantitatively). Then it can readily be verified that Eqs. (12) and (14) have exactly the same proportions between total normal and shear compliances: quantities $\frac{2}{15} N_n / (1 + \gamma)$ and $\frac{2}{5} N_s$ in Eq. (12) and quantities $\frac{4}{15} G_0 Z_n$ and $\frac{2}{5} G_0 Z_s$ in Eq.(14)). Thus representations (11), (12) and (13), (14) are equivalent and differ only by notations.

Assuming that both normal and shear compliances are localized at the same defects (like at penny-shape cracks in conventional models), the ratio $q = N_1 / N_2$ then characterizes the ratio between normal and shear compliances of the crack-like defects. Taking into account the difference in the normal-compliance normalization, one obtains that $\tilde{q} = Z_1 / Z_2 = \frac{1}{2} (N_1 / N_2) / (1 + \nu)$. Comparing Eqs. (11)-(14) with expressions for elastic-moduli reduction based on penny-



shape crack model [9,10], one concludes that penny-shape cracks correspond to the ratio of normal and shear compliances $q = (2 - \gamma)(1 + \nu) \sim 2$ or equivalently $\tilde{q} = (1 - \nu/2) \sim 1$ [12,13].

Since different effective elastic moduli are differently related to normal and shear compliances of the compliant defects, gradual variation of crack density with pressure should correspond to different trajectories of the point $(K(P), G(P))$ on the

$(K, G)$-plane. They are readily expressed via the velocities $V_P$ and $V_s$ of the longitudinal compressional wave (P-wave) and shear-wave velocity (S-wave), which are routinely measured in experiments. (Certainly a different pair of independent moduli can be in principle used.) Comparing the experimentally obtained trajectory with the one theoretically predicted by Eqs. (11)-(14) one can determine the $q$-ratio for real rocks as illustrated in Fig. 3. Such a representation (for example, in $(K, G)$ plane), allows one to exclude intermediate dependences on pressure that in turn dependent on the *a priori* unknown

distributions of the cracks over their aspect ratios. The so-plotted single trajectory makes it possible to reduce the freedom in fitting the two initial experimental curves with additional possibility of scaling pressure axis.

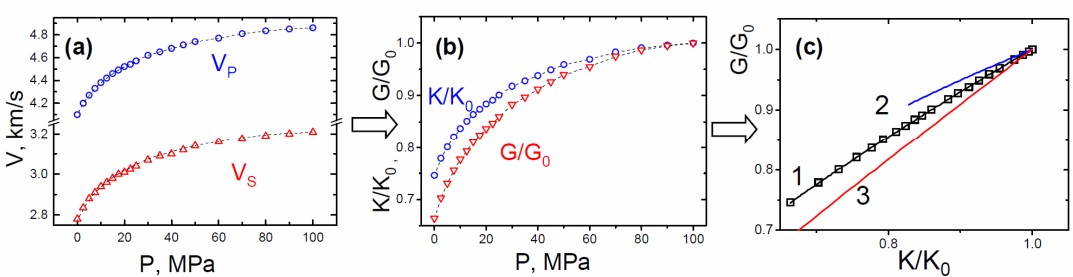

**Fig. 2. Schematic of determining the *q*-ratio of crack compliances via re-plotting the P- and S-wave velocities into**
**trajectory of the point characterizing the rock properties on the (K,G)-plane. (a) - initial pressure-dependences of**
**P- and S-wave velocities. (b) - pressure dependences of the normalized bulk and shear moduli derived from the**
**wave velocities. (c) - the (K,G) plane representing the normalized moduli plotted one against another and**
**superposed theoretical lines with correctly chosen *q*-ratio (curve 1), about 1.5 times overestimated q-ratio (curve 2)**
**and 1.5 times underestimated *q*-ratio. We emphasize that unknown distributions of cracks over their compliance**
**parameters do affect pressure dependencies, but do not affect the so-estimated *q*-ratio.**

This approach was discussed in detail in [14] taking as instructive examples experimental data on pressures dependences (in the range 2-100 MPa) of P- and S-wave velocities for Navajo, Nugget and Weber sandstones used as examples in Fig. 1 [15,19]. The performed examination showed that only for dry Navajo sandstone the q-ratio (appeared to be ~2.35) was more
or less consistent with the expectation $q = (2 - \nu)(1 - \nu) \sim 2.1$ for the model of penny-shape cracks with free faces, whereas for Nugget sandstone it was about twice greater ($q \sim 4.3$) and even grater for Weber sandstone ($q \sim 7 - 8$). The latter





sample even demonstrated negative Poisson's ratio for pressures below 20 MPa. For rocks containing compliant inclusions with dominating normal compliance ( $N_1 > 15\nu + 2(1+\nu)N_2$ ), the presence of negative Poisson's ratio is not surprising [11], see also [27]:

$$\nu_{eff.} = \frac{\nu - \frac{1}{15}N_n + \frac{2}{15}(1+\nu)N_s}{1 + \frac{1}{5}N_n + \frac{4}{15}(1+\nu)N_s} \qquad (15)$$

However, the Weber sandstone containing cracks with significantly increased normal compliance and high concentration of cracks sufficient for making the Poisson's ratio negative looked as a rather exotic example. Similar conclusions on the possibility of negative Poisson's ratio are known for granular materials, in which inter-grain contacts are characterized by normal compliance significantly dominating over the shear one. However, traditionally, negative Poisson's ratios are considered as rather exotic cases mostly for various artificial microstructured solids [28,29].

In what follows we present results of examination of over 90 rock samples, for which data on pressure dependences of P- and S-wave velocities were taken from [15,18,19]. Figure 3 shows the histogram for the Poisson's ratio calculated from the P- and S-wave velocities at the lowest pressure (typically, the available low-pressure data were reported for pressures of several MPa, so that evidently for even lower pressures, crack concentrations were even greater). In this examination we did not try to specially find some specific examples, nevertheless, about 45% of cracked rocks exhibiting pronounced pressure

dependences of the elastic-wave velocities demonstrated negative Poisson's ratio in a few (or at least one) lower-pressure points, where the crack concentration was maximal. Typically the lowest pressures were several MPa and maximal pressures were in the range 50-120 MPa.

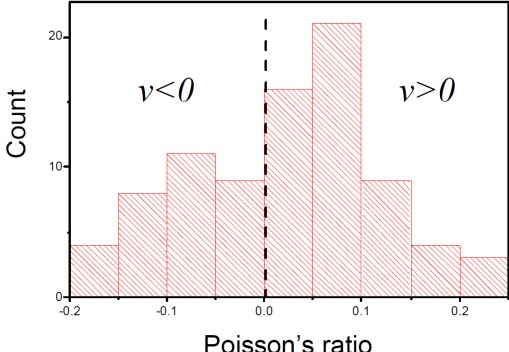

**Figure 3. Histogram for the Poisson's ratios calculated using P- and S-wave velocities for over 90 rocks [15,18,19].**

**Data for minimal pressures (mostly about 8MPa for the available data sets) were used in these calculations. Increasing pressures led to decrease in crack concentration, so that for all rocks the Poisson's ratios gradually became positive.**



For those samples, the initial pressure dependences of the P- and S-wave velocities were re-plotted in the plane of the normalized moduli $(\tilde{K},\tilde{G})$ and the resultant curves were fitted by Eq. (11) and (12) in order to determine the ratio of the compliances of the cracks assuming that the trajectory can be described by a constant $q$-ratio, $q = N_1 / N_2 = const$. This approximation is not *a priory* evident at all, but looks fairly reasonable since the conventional penny-shape cracks indeed

have the $q$-ratio independent of the aspect ratio and, therefore, independent of the pressure of opening/closing of such cracks. For a significant portion of the considered rock samples, the pressure-induced variations for the elastic moduli in the $(\tilde{K},\tilde{G})$ appeared to be surprisingly well described using the approximation of constant $q$-ratio.

It was also found that for two tens of samples, the trajectories could be fairly well fitted by a constant-$q$ curve at higher pressures, but noticeably deviated at lower pressures, usually exhibiting trend characteristic for increasing $q$-ratio. Such

deviations occurred for samples with both negative and positive Poisson's ratios at low pressures. Therefore, for the moment, in the histograms shown below to characterize the revealed $q$-ratios we excluded those samples and retained 71 samples with fairly constant $q$-ratio.

Figure 4 shows the histogram for distribution over $q$-ratio among those 71 samples with $q \approx const$. A striking feature of this histogram is that only the leftmost column (only ~20% of total number of samples) corresponds to $q \sim 2$ in notations [11]

(or $\tilde{q} \sim 1$ in notations [12,13]) that is typical of penny-shape cracks and similar conventionally used crack models. Among the 71 samples presented in Fig. 4 almost one-half (~48%) exhibits negative Poisson's ratio for maximal crack densities at low pressures.

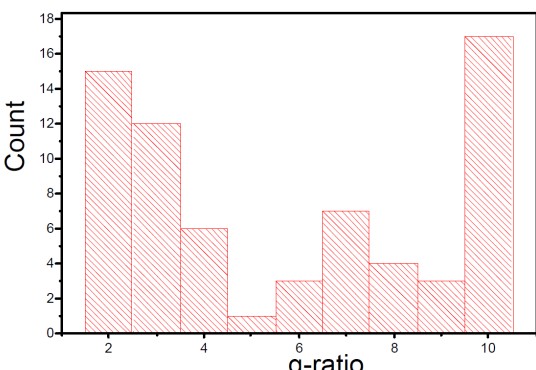

**Fig. 4. Distribution over q-ratio for 71 samples with fairly constant $q$ within the entire pressure ranges including**

**rocks with always positive Poisson's ratio together with samples demonstrating negative Poisson's ratio at lower**

**pressures. The last column includes all samples with $q \geq 10$.**





Figure 5 shows histograms similar to Fig. 4, but separately for 34 samples demonstrating negative Poisson's ratio at low pressures and 37 samples with positive Poisson's ratio in the entire pressure range. As expected from the above-presented arguments (see discussion of Eq. (15)), the $q$-ratios for samples with always positive Poisson's ratio demonstrate the distribution shifted towards small $q$-ratios (Fig. 5b), whereas for samples with negative Poisson's ratio this distribution is

clearly shifted towards high $q$-ratios, significantly higher than $q = (2-\gamma)(1+\gamma) \sim 2$ (or $\tilde{q} \sim 1$) typical of penny-shape cracks (Fig. 5a).

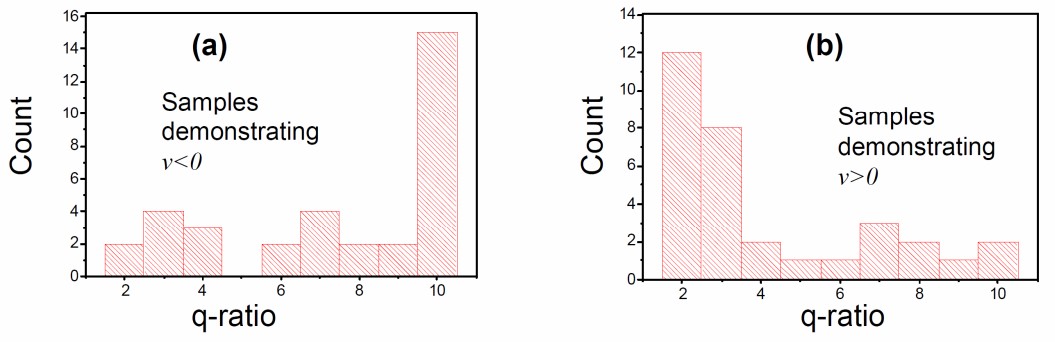

**Fig. 5. Distributions over q-ratio plotted separately for samples from Fig. 4 exhibiting positive and negative Poisson's ratios: (a) the histogram for 37 samples with negative Poisson's ratios at lower pressures; (b) case of 35 samples with**

**always positive Poisson's ratio. The last column in both panels includes all samples with $q \geq 10$.**

It should be mentioned that increased $q$-ratio was also found in the case of samples with always-positive Poisson's ratio (as can be seen in Fig. 5b). However, by applying procedures shown in Fig. 2 we verified that for these samples, the crack density is significantly smaller than for the rocks exhibiting negative Poisson's ratio. Namely, for rocks with negative

Poisson's ratio and $q \sim 5-10$ or even greater, typically the crack density is $\nu_0^{(\varsigma)} \sim 1-2$, whereas for rocks with similar increased $q$-factor, but positive Poisson's ratio even at lowest pressures, the crack density is significantly lower $\nu_0^{(\varsigma)} \sim 0.1-0.2$.

## 4 Conclusions

In the described analysis of pressure-dependent (i.e. nonlinear) elastic rocks' properties we used approaches [11,14] and

[12,13] in which the effective-medium model is based on generalized phenomenological representation of cracks as highly compliant defects whose compliance properties are not *a priori* predetermined, so that the proportion between the normal and shear compliances and their integral amounts can be found from the comparison with experimental data. It should be





emphasized that such comparison is essentially based on the usage of numerous data points obtained in fairly wide range of pressures. This consideration of the large data sets describing nonlinear behaviour of rocks ensures much better reliability and accuracy than comparison of a pair of points (e.g. for two pressure levels).

The performed examination has indicated that properties of compliant cracks in many rocks reasonably well agree with the

assumption about uniform distribution $v_0^{(\varsigma)} \approx const.$ of the cracks over their compliance parameter (i.e. actually their aspect ratio), which gives a simple way (actually a single parameter $v_0^{(\varsigma)}$) for comparison of crack concentrations in different samples.

The usage of the theoretical description [11,12,13,14] with explicitly introduced normal and shear compliances of the defects made it possible to determine this ratio for real cracks from the trajectory of $(K(P), G(P))$ in the $(K, G)$ plane. Using the

literature data on pressure dependences of P- and S-wave velocities [15,18,19], about 90 rock samples were examined. For a significant portion (~80%) of the samples the $q$-ratio between the normal and shear compliances appeared to be significantly different from what would be predicted by the conventional crack models. These observations agree with some other results based on smaller volumes of data [14,16], which also indicates that quite often the conventionally used crack models (like the penny-shape one) cannot adequately describe properties of real rocks. In fact for the considered 71 samples that can be

well described in the approximation of constant $q$-ratio, it appears that only ~20% of rocks demonstrate $q \sim 2$ typical of the penny-shape cracks.

Furthermore, the performed examination of pressure dependences for ~90 samples (found in literature without any special selection) revealed that a significant portion of samples (about 45%) demonstrated negative Poisson's ratio at low pressures, for which concentrations of open cracks were maximal. Such a significant number of naturally auxetic rocks is in contrast

with the conventional viewpoint that occurrence of negative Poisson's ratio for rocks is an exotic fact [28,30]. Previously mainly artificial materials with the microstructure engineered to exhibit negative Poisson's ratio (auxetic materials) were discussed in the literature (see e.g., reviews in [29], [31]).

The performed comparison of $q$-ratios has shown that for samples exhibiting negative Poisson's ratio, the distribution of determined compliance ratios for cracks shows clear distortion towards large $q$-ratios (strongly dominating normal

compliance of cracks over their shear compliance). This finding perfectly agrees with theoretical models for crack-containing solids and granular materials, according to which negative Poisson's ratio can be obtained in nearly isotropic material only if the cracks or contacts have dominating normal compliances [11,27]. In contrast, for samples with positive Poisson's ratio, the determined distributions of $q$-ratios demonstrated a clear distortion towards small values.

Overall, the obtained results indicate the necessity of further development of crack models to account the revealed numerous

examples of rocks with defects demonstrating $q$-ratios significantly greater than for penny-shape cracks and similar conventionally used crack models.





**Acknowledgement**. VYZ and AVR acknowledge the support of the Russian Foundation for Basic research (grant No 15-05-05143). AVD and EP acknowledge the financial support from the Australian Research Council linkage project LP 120200797, Australian Worldwide Exploration (AWE) limited and Norwest Energy NL Companies.

Guyer, R.A. and Johnson P.A.: Nonlinear mesoscopic elasticity: the complex behaviour of rocks, soil, concrete. John Wiley & Sons, 2009.

Zaitsev, V.Yu., Gurbatov, S.N., and Pronchatov-Rubtsov, N.V.: *Nonlinear Acoustic Phenomena in Structure-Heterogeneous Media* (In Russian), Editions of the Institute of Applied Physics RAS, Nizhny Novgorod, 2009.

Zaitsev, V.Yu.: A model of anomalous acoustic nonlinearity of micro-inhomogeneous media, Acoust. Lett., 19, 171-176, 1996.

Belyaeva, I.Yu. and Zaitsev, V.Yu.: Nonlinear Elastic Properties of Microinhomogeneous Hierarchically Structured Media, Acoust. Phys., 43, 510-515, 1997.

Belyaeva, I. Y. and Zaitsev, V. Yu.: The limiting value of the parameter of elastic nonlinearity in structurally inhomogeneous media, Acoust. Phys., 44, 635-640, 1998.

Zaitsev, V.Yu. and Matveev, L.A., Strain-amplitude dependent dissipation in linearly dissipative and nonlinear elastic microinhomogeneous media, Russian Geology and Geophysics, 47, 694-709, 2006.

Zaitsev, V.Y., Saltykov, V.A., and Matveev, L.A.: Relation between the tidal modulation of seismic noise and the amplitude-dependent loss in rock, Acoust. Phys., *54*, 538–544, 2008. http://doi.org/10.1134/S1063771008040143

Walsh, J. B.: The effect of cracks in rocks on Poisson's ratio, J. Geophys. Res., *70*, 20, 5249-5257, 1965.

O'Connell, R.J. and Budiansky, B.: Seismic velocities in dry and saturated cracked solids, J. Geophys. Res., 79, 5412-5426, 1974.

Zimmerman, R.W.: The effect of microcracks on the elastic moduli of brittle materials, J. Mat. Sci. Lett., 4, 1457–1460, 1985.

Zaitsev, V. and Sas, P.: Elastic Moduli and Dissipative Properties of Microinhomogeneous Solids with Isotropically Oriented Defects, Acta Acustica United with Acustica, 86, 216–228, 2000.

MacBeth, C.: A classification for the pressure-sensitivity properties of a sandstone rock frame, Geophysics, 69, 497-510, 2004.

Sayers, C. M. and Kachanov, M.: Microcrack induced elastic wave anisotropy of brittle rocks, J. Geophys. Res., 100, 4149–4156, 1995.

Zaitsev, V. and Sas, P.: Effect of the high-compliant porosity on variations on P-and S-wave velocities in dry and saturated rocks: comparison between theory and experiment, Phys. Mesomech., 7, 37-48, 2004.

Coyner, K.B.: Effects of stress, pore pressure, and pore fluids on bulk strain, velocity, and permeability in rocks, Ph.D. thesis, Massachusetts Institute of Technology, 1984.



Gurevich, B., Makarynska, D., and Pervukhina, M.: Are penny-shaped cracks a good model for compliant porosity? In SEG Technical Program Expanded Abstracts. Society of Exploration Geophysicists, 3431-3435, 2009.

Sayers, C.M. and Han, D.-H.: The effect of pore fluid on the stress-dependent elastic wave velocities in sandstones, In 72nd Annual International Meeting, SEG, Expanded Abstracts, 1842–1845, 2002.

Freund, D. Ultrasonic compressional and shear velocities in dry clastic rocks as a function of porosity, clay content, and confining pressure, Geophys. J. Int., 108, 125-135, 1992.

Mavko, G. and Jizba, D.: The relation between seismic P- and S-wave velocity dispersion in saturated rocks, Geophysics, 59, 87–92, 1994.

Han, D.: Effects of porosity and clay content on acoustic properties of sandstones and unconsolidated sediments: Ph.D. dissertation, Stanford University, 1986.

Zarembo, LK, Krasilnikov, VA: Nonlinear phenomena in the propagation of elastic waves in solids. Soviet Physics Uspekhi 13, 778–797, 1971.

Trivett, D. H., Pincon, H., and Rogers, P. H.: Investigation of a three-phase medium with a negative parameter of nonlinearity. *J. Acoust. Soc. Am.*, *119*(6), 3610-3617, 2006.

Zaitsev, V. Y., Dyskin, A., Pasternak, E., & Matveev, L.: Microstructure-induced giant elastic nonlinearity of threshold origin: Mechanism and experimental demonstration. *Europhys. Lett.*, *86*(4), 44005, 2009. http://doi.org/10.1209/0295-5075/86/44005

Morlier, P.: Description de l'etat de fissuration d'une roche a partir d'essais non-destructifs simples, Rock mechanics, *3*, 125-138, 1971.

Mavko, G. M. and Nur, A.: The effect of nonelliptical cracks on the compressibility of rocks, J. Geophys. Res.: Solid Earth, 83, 4459-4468, 1978.

Landau LD, Lifshitz EM.: *Theory of Elasticity*. Oxford: Butterworth-Heinemann; 1986.

Pasternak, E. and Dyskin A.V.: Materials and structures with macroscopic negative Poisson's ratio. *Intern. J. Engineering Sci.*, 52, 103-114, 2012

Gercek, H. Ã.: Poisson's ratio values for rocks. *Int. J. Rock Mechanics and Mining Sciences*, *44*, 1–13, 2007. http://doi.org/10.1016/j.ijrmms.2006.04.011

Lakes, R.: Advances in negative Poisson's ratio materials, Advanced Materials, 5, 293-296, 1993.

Jizba, D.L. Mechanical and acoustical properties of sandstones and shales. Diss. to the Department of Geophysics. Stanford University, 1991.

Shufrin, I., E. Pasternak and A.V. Dyskin: Negative Poisson's ratio in hollow sphere materials. *International Journal of Solids and Structures*, 54, 192-214, 2015.