# Peer review of "Extracting real-crack properties from nonlinear elastic behavior of rocks: abundance of cracks with dominating normal compliance and rocks with negative Poisson's ratio"

_Nonlinear Processes in Geophysics, 2017_

## Referee Comment (RC1) · Anonymous Referee #1 · 22 May 2017

Examined article is theoretical and considers the approach for description of the important and interest problem connected with determination of the behaviour of the solid medium with distributed defects (cracks). In contrast to works in which the penny-shape cracks and thin elliptical voids with small aspect rations have been used individually, in the examined article the new approach bases on the examination of the aggregate of the cracks of two types. The cracks are considered as a highly compliant defects with independed normal and shear compliances not restricted by a predetermined proportion between them.

[Figure]

Authors propose an approach to the description of the deformation of mediums with compliant defects using introduced parameter characterized the relative compliance of the defects, and demonstrate possibilities of sufficiently strong variations of tensosensitivity also elastic modulus from the stress, and (the essentials) – possibility existence of negative Poisson's rations under the certain conditions. The authors collected published data and showed that experiments demonstrate negative Poisson's ratios in a number of cases.

Results of the examined work in spite of some singularity and absent of the clear substantiation of the physical mechanisms resulting to sufficiently high negative Poisson's ratios are of interest as a whole and could be carried to scientific community. Publication of this article will be used for good basis for discussions and attracting attention to examined problem.

---

## Referee Comment (RC2) · Anonymous Referee #2 · 26 May 2017

This paper discusses a relevant and interesting problem concerning the effect of pressure on the elastic moduli of rocks with a high concentration of cracks. The basic analysis is based on the study of elastic wave velocities at different pressures. A large amount of experimental data has been analyzed. An approach is used in which an effective medium with pressure sensitive elastic moduli is considered without regard for the specific crack shape and crack orientation. The examples provided indicate significant changes in the elastic wave velocities and corresponding elastic moduli. An important point is that independent relations are allowed for in compression and shear models. An unexpected result is a large amount of data on the auxetic properties of

rocks for which negative Poisson's ratios have been obtained. The reported results raise some nontrivial questions concerning the Poisson's ratio determination in terms of the elastic wave velocities in fractured and heterogeneous media. Moreover, what are the characteristics of this parameter in fractured media? I would recommend to continue this research and to study the effect of pressure not only on dynamic but also on static moduli. This paper is of considerable interest and can be published as presented.

Please also note the supplement to this comment:
http://www.nonlin-processes-geophys-discuss.net/npg-2017-9/npg-2017-9-RC2-supplement.pdf

---

## Author Comment (AC1) · 6 Jul 2017

We thank the referees for positive comments.

---

## Author Response (AR1)

We thank the referees for their positive opinion about our manuscript. Since both referees did not require any specific modifications and even recommended the manuscript acceptance in its initial form, in the present version we made only minor corrections of a few misprints